# Co-Graft of Acellular Dermal Matrix and Split Thickness Skin Graft—A New Reconstructive Surgical Method in the Treatment of Hidradenitis Suppurativa

**DOI:** 10.3390/bioengineering9080389

**Published:** 2022-08-14

**Authors:** Marcin Gierek, Wojciech Łabuś, Anna Słaboń, Karolina Ziółkowska, Gabriela Ochała-Gierek, Diana Kitala, Karol Szyluk, Paweł Niemiec

**Affiliations:** 1Center for Burns Treatment, ul. Jana Pawła II 2, 41-100 Siemianowice Śląskie, Poland; 2Dermatology Department, City Hospital in Sosnowiec, ul. Zegadłowicza 3, 41-200 Sosnowiec, Poland; 3Department of Physiotherapy, Faculty of Health Sciences in Katowice, Medical University of Silesia in Katowice, 40-752 Katowice, Poland; 4I Department of Orthopaedic and Trauma Surgery, Ortophaedics Department, District Hospital of Orthopaedics and Trauma Surgery, 41-940 Piekary Śląskie, Poland; 5Department of Biochemistry and Medical Genetics, Faculty of Health Sciences in Katowice, Medical University of Silesia in Katowice, 40-752 Katowice, Poland

**Keywords:** acellular dermal matrix, hidradenitis suppurativa, surgical treatment, split thickness skin graft

## Abstract

Hidradenitis suppurativa is a chronic disease that significantly reduces patients’ quality of life. Patients are chronically treated with systemic therapies, which are often ineffective. Surgical treatment for severe cases of hidradenitis suppurativa is one option for affected patients. Surgical treatment has its limitations, and wound closure may be particularly problematic. This requires the use of reconstructive techniques. The methods of choice for wound closure are split-thickness skin grafts or local flaps reconstructions. However, each method has its limitations. This is a presentation of a new reconstructive surgical method in hidradenitis suppurativa surgery: the use of a co-graft of Acellular dermal matrix and split thickness skin graft as a novel method in wound closure after wide excisions, based on two cases. The results of this method are very promising: we achieved very fast wound closure with good aesthetic results regarding scar formation. In this paper, we used several examinations: laser speckle analysis, cutometer tests, and health-related quality of life (QoL) questionnaire to check the clinical impact of this method. Our initial results are very encouraging. ADM with STSG as a co-graft could be widely used in reconstructive surgery. This is a preliminary study, which should be continued in further, extended research.

## 1. Introduction

Hidradenitis suppurativa (HS), also known as Acne Inversa, is an inflammatory skin disease with a clinical presentation of characteristic recurrent, chronic, painful and suppurating lesions [1,2]. HS was first described by Alfred ALM Velpeau, a French surgeon, in 1839 [3]. HS has a devastating impact on quality of life. Patients suffer from social stigma, poor mental health and higher suicide rates compared with the general public. Hidradenitis suppurativa has a significant impact on health-related quality of life. HS has mental and physical effects. Pain, discharge, smell and pruritus are the most common symptoms, which entirely change patients’ lives [4]. HS has been documented to have a negative influence on patients’ health-related quality of life [4]. Moreover, the disease is often correlated with severe consequences, such as a higher incidence of depression, the fear of stigmatization and suicidal tendencies [4].

The global prevalence of HS has been reported in the 0.00033–4.1% range [5,6]. The exact etiology and pathogenesis of HS remains unclear [7,8]. The cause of follicular plugging is still under debate, although immune dysregulation, genetic factors, hormonal fluctuation, and environmental risk factors are thought to play a role [7]. The most common sites affected by HS are: axillary, inguinoperineal, gluteal, submammary regions and the nape of the neck, waistband, and inner thighs [9]. HS often has a delayed diagnosis. An average delay of 7–10 years has been reported between disease onset and diagnosis [4].

HS has mild, moderate and severe courses. Medical treatment with topical or systemic antibiotics is widely used as a first-line treatment in the management of mild disease at early stages. In chronic (Hurley II/III) cases, surgical methods should be considered [10,11]. Health-related quality of life significantly improves after surgical treatment [4]. The most important aspect of surgical treatment of HS is wide excision. Afterwards, the most problematic aspect is wound closure, due to the wide excision (very large wounds, e.g., bottocks). In such cases, the use of reconstructive techniques is required, such as skin flaps, skin grafts or both [10,11]. The surgical reconstructive techniques for HS are limitless. There is no perfect method of wound closure in HS surgery. Skin flaps appear to be the best method regarding tissue coverage; however, they may cause the displacement of potentially diseased tissues, which may lead to HS recurrence. Skin flaps are associated with larger scars and the possibility of vascular complications such as skin flap necrosis. Split thickness skin grafts may cause inelastic contraction scars [10,11]. With any surgical wound, the main goal is rapid healing with a minimal scar that will have a long-term natural contour and provide lasting tensile strength. Wound healing ideally begins with primary wound closure. The human body conducts the natural substrates for healing and offers the best chance for positive results regarding healing. Sometimes, tension is too great for primary closure, and additional tissue is needed to complete the repair [12]. The acellular dermal matrix (ADM) is a class of biological, synthetic, and composite scaffold materials used to augment and replace deficient or missing skin and soft tissues [13]. ADM is made by taking a full-thickness section of skin from a donor source—which, in most cases, is a human cadaver, porcine, or bovine in origin. The tissue is run through a series of steps (enzymatic decellularization) to eliminate all cellular elements from the tissue; as a result, we reached an acellular matrix made by collagen. [12]. For ADMs to be successfully applied in clinical applications, they must allow for host vascular ingrowth through angiogenesis, where host blood vessels grow into the empty spaces left after ADM. With vascular ingrowth, the ADM is said to “integrate” into the patient and become live connective tissue. The initial uses for ADMs were to augment skin replacement therapies in burn patients. Soon afterwards, the use of ADMs evolved into other, varied applications [13].

There are reports of new acellular dermal matrix (ADM) applications as a co-graft with split thickness skin graft (STSG), with good healing effects, and the very elastic formation of scars [14].

Due to these imperfections in the current surgical methods, there is a need to search for a new, better method. We would like to present a novel method of surgical treatment in HS (co-graft of Acellular Dermal Matrix and Split Thickness Skin Graft), assisted by laser speckle analysis, based on two case reports of patients treated in a Center for Burns Treatment in Siemianowice Slaskie, Poland. We would like to introduce the world’s first two successful reconstructions with ADM and STSG (co-graft) in hidradenitis suppurativa surgery.

## 2. Materials and Methods

The study was conducted in accordance with the Declaration of Helsinki, and approved by the Institutional Ethics Committee of Silesian Medical Chamber in Katowice (protocol code 30/2021 and 22 September 2021 date of approval). The study was also approved by Institutional Review Board in Center for Burns Treatment (code 2/2022/CLO) as a clinical trial. All patients gave informed consent to share their medical data and images.

### 2.1. First Case

The first presented patient is a 31-year-old man with a 13-year history of hidradenitis suppurativa. Changes occurred in the armpits, groin, buttocks. Hurley Score-III. Co-morbidities: condition after hip surgery due to Perthes disease (1998); apart from HS, there were no additional diseases. He has a history of smoking. He came to the Center for Burns Treatment in Siemianowice Slaskie, Poland for the surgical treatment of purulent lesions in his buttocks. The patient qualified for surgical treatment, but due to the extensive lesions, affecting 75% of the buttocks, a decision was made to perform 2-stage surgical treatment. Hospitalization in the general surgery department lasted 16 days. (Figure 1).

In the first stage of treatment, lesions covering the buttocks were excised, two local flaps were moved to partly close the wound and local negative pressure therapy (Vacuum-Assisted Closure (VAC)) was applied. After 5 days of VAC therapy, another surgery was decided. The patient was qualified to have the buttocks wounds closed with an ADM and STSG co-graft. Before the operation, after the operation, and during each dressing change in the hospital, the patient was subjected to speckle laser examinations (Laser Speckle Contrast Analysis (LASCA)) to assess tissue perfusion. Through this, we were able to monitor the healing process. After 16 days of hospitalization in the Center for Burns Treatment in Siemianowice Slaskie, the patient was discharged. The patient continued to be treated at the outpatient surgery clinic, where he came for regular visits (follow up 2.5 months after surgery). At each visit (during healing), speckle laser (LASCA) microperfusion was assessed, and on the last 2 visits, the patient’s skin was examined with a cutometer and skin USG scanner. After 2 months of treatment, the patient was completely healed, with a very good aesthetic effect of the scars. He is still under the constant control of our team.

### 2.2. Second Case

Patient, male, 36 years old, suffered from hidradenitis suppurativa for 15 years. He had been treated many times by dermatologists and surgeons. The lesions affected both armpits (Figure 2). Lesions Hurley score-III. Patient with BMI > 35, non-smoker. The patient was not chronically ill with any other diseases. The patient came to the Center for Burns Treatment in Siemianowice Slaskie for surgical treatment of the right armpit. During hospitalization, due to the extensive lesions, a decision was made to use the ADM and STSG co-grafts, and local flaps. The patient was operated on once and discharged after 9 days of hospitalization. Before surgery, after surgery and at each dressing change, the patient was assessed for microperfusion by laser speckle analysis (LASCA). The patient was further treated in the outpatient surgery clinic, where he also had LASCA microperfusion assessed at each visit and was examined with a cutometer and skin USG scanner on the last 2 visits. After 2 months of treatment, the patient was completely healed, with perfect elastic scars. The skin is very elastic, and, moreover, during scar formation there was no noticeable characteristic meshing, as is typical with STSG scars. The patient is still under the constant control of our team.

### 2.3. Surgical Technique-Co-Graft of ADM and STSG

We did not use commercial ADM in our operations. The ADM we used was our own in-house ADM, obtained from a cadaver as an allograft and then enzymatically processed in our Tissue Bank laboratory in Center for Burns Treatment in Siemianowice Slaskie, Poland. The samples of human skin (intended for ADMs preparation) were collected from multi-organ/multi-tissue deceased human donors (the donor sites included such areas as arms, forearms, thighs, lower legs, calves, chest, bottocks and back). The battery dermatome (Aesculan III, Aesculap AG, Tuttlingen, Germany) was used. Skin depth was set to parameter 4–5 on the dermatome scale (0–14 scale).

All activities concerning storage, transportation to the tissue bank and the preparation of allogeneic human skin grafts were carried out in accordance with the relevant standard operating procedures (SOPs) of the Tissue Bank, in the Center for Burns Treatment in Siemianowice Slaskie, Poland. In brief, the tissue engineering (preparation) of allogeneic human Acellular Dermal Matrix (ADM) consists of rinsing the skin grafts in hypotonic solution, separation of the dermis from the epidermis (1–3 h of incubation in 2.4 u/ml Dispase II, Gibco/Thermo Fisher Scientific, New York, NY, USA) and 24 h incubation of human dermis in 0.05% trypsin solution in EDTA (Life Technologies, Carlsbad, CA, USA); the enzyme solution was changed after 12 h incubation. After the decellularization process, the uneven fragments of the ADMs were cut off and the graft area was measured. Then, each graft was registered in the ISBT 128 computer system and packed into triple bags. ADMs were sterilized by electron beam radiation dose of 35 kGy. The sterility of the ADMs was confirmed by microbial culture. Donors were HIV, HCV and Hepatitis B negative (which was confirmed before the donation).

The ADM obtained this way was transplanted into the wounds. In the first case, after removal of VAC therapy, we covered the wounds with a ratio 1: 1.5 mesh ADM, and in the second case, we used a ratio 1: 2.0 mesh ADM, transplanted directly onto the excision wound. In the second stage of each surgical procedure, after ADM fixation, we collected the skin for grafting from the thighs in a ratio of 1: 1.5 meshed (in Case 1) and 1:2.0 (in Case 2). We grafted the STSG directly onto the ADM layer (Figure 3 and Figure 4).

### 2.4. Examinations

Laser Speckle Contrast Analysis (LASCA)

Laser speckle contrast analysis (LASCA), also known as laser speckle contrast imaging (LSCI), is a method that instantly visualizes microcirculatory tissue blood perfusion. It is an imaging technique that combines high resolution and high speed. When an object is illuminated by laser light, the backscattered light will form an interference pattern consisting of dark and bright areas. This pattern is called a speckle pattern. If the illuminated object is static, the speckle pattern is stationary. When there is movement in the object, such as red blood cells in a tissue, the speckle pattern will change over time. A camera with a fixed exposure time will record these changes in the speckle pattern as motion blurring. The LASCA examination was performed on specific days: before admission, immediately before the procedure, immediately after the procedure and in days following the procedure. The measuring device was placed 30 cm from the test object. The examination lasted 20 s, with an image width 20.1 cm, height 19.0 cm; 16 frames, 1 color photo per 10 s, and resolution 0.78 mm. All data on LASCA device collected are in Figure 5, Figure 6, Figure 7 and Figure 8. The region of interest (ROI) was drawn during the first measurement, and this ROI accompanied all subsequent measurements. An ROI 1 was established, which referred to the entire wound/skin lesion; ROI referred to the most bloodshot area and ROI 3 referred to healthy skin and was taken as a reference (Figure 6 and Figure 8).

Cutometer^®^ dual MPA 580, Courage + Khazaka electronic GmbH, Mathias-Bruggen 50829

Temperature measurements

The measurement was carried out using a special lens and an IR; the detector measured the level of infrared radiation emitted by the skin. The skin temperature was determined by microcirculation. The weaker the circulation in a given area of the skin, the lower the temperature (Cutometer^®^ dual MPA 580, Courage + Khazaka electronic GmbH, Mathias-Bruggen 50829 Köln/Germany).

Corneometer measurments

The measurement of skin moisture was based on different dielectric constants of water (81) and other substances (most often <7). The measuring capacitor shows the changes in capacitance according to the moisture content of the samples. A glass plate separated the metal (gold) paths inside the head from the skin to prevent current conduction in the sample. An electric field was created with variable attraction between paths. One path had a surplus of electrons (negative sign), and the other had a shortage of electrons (positive sign). During measurements, the dispersion field penetrated the first layer of the skin and the capacity was determined.

Mexameter measurments

The measurement of skin color was based on the principle of absorption. The Mexameter^®^ MX 18 probe emits light at three specific wavelengths. The receiver measures the light reflected by the skin. The positions of the transmitter and receiver ensured that only scattered light was measured. Since the amount of light emitted is predetermined, it was possible to calculate the light absorbed by the skin.

Tewameter measurments

The tewameter probe indirectly measured the density gradient of water evaporated from the skin. This was completed with the help of two sensors (temperature sensor and relative humidity sensor) inside the empty cylinder. This was an open-chamber measurement. Measurement in an open chamber is the only measurement method that does not affect the continuity of the epidermis. The basis of the measurement is the law of diffusion.

Cutometer measurments

A cutometer was used to measure the upper layer of the skin with the help of a vacuum, which caused mechanical deformations of the skin. The measurement was based on the suction method. The device creates a vacuum that pulls the skin into the slot of the probe, and then releases it after a certain period of time. Inside the probe, using a non-contact optical measuring system, the penetration depth is measured. The optical measurement system consists of a light source and a light receiver, as well as two prisms that are directed towards each other, which cast light from the transmitter to the receiver. The intensity of light varies depending on the penetration depth of the skin. The resistance of the skin to the vacuum and its ability to return to its original shape are displayed as curves.

Skin Ultrasound DUB SkinScanner v5.31 Dubai Silicon Oasis, Dubai UAE

This is a high-frequency and high-resolution digital ultrasound imaging system for the skin. Presentation B (Brightness) visualizes a two-dimensional cross-section, in which the instantaneous value of the received signal modulates the brightness of subsequent points of the image.

## 3. Results

At the very beginning, after the dressings were unwrapped, delayed healing was noticeable. After unwrapping the dressings, we noticed pale STSGs in both cases. In Case 1, we used ADM meshing in ratio 1:1.5, and in the second operation we changed the ratio to 1:2.0. We changed the ratio due to our first observations that ratio 1:1.5 co-grafts were pale for the first 48 h. For the second case, a few days after first surgery, we decided to change this ratio to 1:2.0. In our opinion, meshing in a ratio of 1:2.0 accelerates vascularization; we observed better co-graft vital signs in this ratio (physical examination each day during dressing changes). We performed a speckle laser examination LASCA (Perimed AB, Stockholm, Sweden) to assess microperfusion, which, with each subsequent day of the examination, showed more and more hyperemia, indicating healing and vascularization (Figure 5 and Figure 7). This delayed vascularization is due to the fact that ADM is vascularized first, and then STSG. During visits to the outpatient surgery clinic, we also performed LASCA examinations, according to which we monitored the entire process of vascularization (Figure 5, Figure 6, Figure 7 and Figure 8).

At the end of the healing process (follow up for a 2.5-month period), we performed a cutometer (Cutometer^®^ dual MPA 580, Courage + Khazaka electronic GmbH, Köln, Germany) and tested parameters such as temperature, corneometer, mexameter, and TEWL. All data are attached in the Appendix A).

Additionally, we performed a skin ultrasound (DUB SkinScanner v5.31, Dubai Silicon Oasis, Dubai, UAE) to assess ADM and STSG thickness compared to local skin flaps and compared to healthy skin. These studies showed that, using the ADM + STSG method, the thickness of the epidermis was 135 μm, the epidermal density was 23.07% and thickness of the dermis was 1162 μm at 2.5 months after the surgical procedure (all data are available in the Appendix A). On physical examination, a very good, flexible scar was noticeable in both cases, as shown in Figure 9. Patients did not complain of pain in the area of the scar. We also used the EQ-5D-5L health-related quality of life questionnaire (EuroQol Group EQ-5D™, Rotterdam, The Netherlands), which showed that patients’ quality of life improved in both cases. Health-related QoL in Case 1, before surgery, was 4 points (maximum 100 points), and in follow-up 2.5 months after procedure, this reached 86 points (maximum 100 points), showing an 82-point improvement in health-related QoL. In case 2, QoL before surgery was 10 points (maximum 100 points) and after the procedure, at 2.5 months follow-up, this reached 100 points (maximum 100 points), with a 90-point improvement in QoL.

These observations showed that post-surgical health-related QoL is improved by the co-graft ADM and STSG method. The preliminary results of our observations are very promising; scar formation is very elastic with satisfying aesthetic effect (Figure 9).

## 4. Discussion

In our work, we presented the first two cases of co-graft ADM and STSG as a new method of surgical treatment for hidradenitis suppurativa. This method is innovative and improves the reconstructive techniques that are available for large skin defects after excision. This method may give lead to treatment results, which will significantly improve the speed of healing and recovery.

The skin is a collagen-rich tissue and is a rich source of the biomaterials used in tissue engineering [14,15]. By removing cells from allogeneic human dermis, a cell-free, collagen, non-immunogenic scaffold is obtained, which can be revitalized de novo by autologous cells. Such an acellular dermal matrix (ADM) or acellular dermal graft (ADG) can be a stimulus for natural mechanisms of regeneration and reconstruction [14,15,16,17,18,19,20,21,22]. The immune response is mainly directed against proteins and lipids of the cell membrane; therefore, the extraction of cells from the tissue is a promising way to avoid the emergence of a post-transplant immune response in the recipient organism [23,24,25]. It was found that collagen, with slight interspecies differences among the type I collagen family, is clinically characterized by low immunogenicity [26]. The low immunogenicity of type I collagen results from slight differences in the composition of amino acids [27]. Although the clinical use of cell-free tissue engineered dermal matrices (ADM) is a relatively new approach in the history of wound healing, the use of natural elements, such as the acellular extracellular matrix of the skin, is an increasingly common strategy in regenerative medicine and tissue engineering, especially for the management of a burn or chronic wound [28]. As shown by the results of the presented report, the thickness of the epidermis and dermis may depend on the study area. Moreover, the skin thickness may be influenced by factors such as the sex of the examined subject and skin pigmentation [29,30]. The test results may slightly differ depending on the chosen research method [30,31]. The thickness of the epidermis of a healthy person should reach from about 39 μm to 129 μm, and the thickness of the dermis should range between 943 μm and 1941 μm [32]. By comparing our results with these ranges, we can conclude that the ADM and STSG co-grafts fall within these ranges.

The results of the obtained measurements may slightly differ from the norm due to the adopted research method. According to our knowledge of the dimensions of healthy skin, we were able to compare wounds during the healing process. ADM is widely used in breast reconstruction after mastectomy for oncological reasons. This topic is quite well known and adopted in common surgical oncology [33,34,35,36,37]. Haney et al. used ADM in genitourinary reconstructive surgery, which is a new approach for ADM [38]. ADM is also used in burn treatments [39,40,41,42,43,44,45,46].

We found only a few references to ADM and STSG co-grafts in the medical databases. Lee YJ et al. presented great results of ADM and STSG’s synergistic therapeutic effect in the treatment of deep tissue defects in the donor site of free flaps [47]. Lee YJ et al. compared ADM + STSG to STSG alone. Their results showed that the Vancouver Scar Scale, vascularity sub-score (*p* = 0.003) and total score (*p* = 0.016) were significantly lower in the skin graft with the ADM group. Looking at the Patient and Observer Scar Assessment Scale, the pain (*p* = 0.037) and stiffness sub-score (*p* = 0.002), and total score (*p* = 0.017) were found to be significantly lower in the skin graft with the ADM group [47]. Lee et al. concluded that STSG with ADM results in better scar quality in objective and subjective terms, similar to our results. Lee et al. encouraged us to use the co-graft method in hidradenitis suppurativa surgical treatment.

Chaffin et al. presented six cases that used ADM in HS treatment—probably the first use of ADM in the surgical treatment of HS in the literature. In four cases, ADM was followed by skin flap reconstruction. In two cases, ADM was used as a skin substitute. In one case, 22 weeks after ADM application, STSG was used. However, Chaffin et al.’s method did not present an ADM and STSG co-graft [48].

Such a method cannot be found in the literature for the treatment of hidradenitis suppurativa. This work is the first description of this new method in the surgical treatment of hidradenitis suppurativa. ADM and STSG co-grafts provide very good healing effects. Moreover, the new research methodology presented by us, including the use of speckle laser analysis, provides the possibility of very precise monitoring of the healing process. Our two cases were the first two successful reconstructions with ADM and STSG co-grafts in large skin defects after wide excisions of HS. ADM and STSG seem to provide good final results, as well as aesthetic and very flexible scars.

The most important issue in our manuscript is the presentation of a new reconstructive method (ADM and STSG co-graft) in the treatment of large defects after wide excisions in the treatment of hidradenitis suppurativa, which has not been presented to date. ADM and STSG co-grafts can be more widely used in other applications in reconstructive surgery, e.g., after injuries, or in reconstructions after wide oncological excisions, where the possibility of using a rotatory flap is limited.

Additionally, this manuscript presents an in-depth laser speckle contrast analysis (LASCA) study, which, to our knowledge, has never been used before to monitor healing after the surgical treatment of hidradenitis suppurativa, or even when using STSG in other cases. The LASCA microperfusion assessment can also be used in other healing monitoring applications, such as skin flaps, skin grafts, and microsurgery techniques. Our study limitation is the small group of operations (two cases) performed to date. Our follow-up period was only 2.5 month post-operatively, but this is a limited preliminary study. In further research, we will prolong the follow-up. Many HS patients are obese, diabetic, and heavy smokers, which can definitely affect wound healing as well. From our perspective, we observe that STSGs are often not adopted in such patients. Perhaps the simultaneous use of ADM and STSG co-grafts could improve healing in such patients. This also requires further research on the healing process of ADM and STSG in comorbidities.

The other limitation of this method is the price. ADM is quite expensive (commercial ADM), and using in-house (as in our method) ADM is not possible in every plastic surgery/general surgery department.

In our Center, there is a tissue bank with a tissue laboratory, where all the complicated ADM manufacturing processes take place. Further research into ADM and STSG should be conducted.

## 5. Conclusions

The surgical treatment of large skin defects, as in the case of HS lesions, requires the use of modern reconstructive methods. Skin flaps and split thickness skin grafts are proven methods, but have limitations. It is to early to present the long-term effects of this treatment, but first preliminary results are very encouraging. Using a co-graft of ADM and STSG as the method for final wound closure in hidradenitis suppurativa surgery seems to be a promising improvement to existing techniques, but requires further research and long-term follow-up.

## Figures and Tables

**Figure 1 bioengineering-09-00389-f001:**
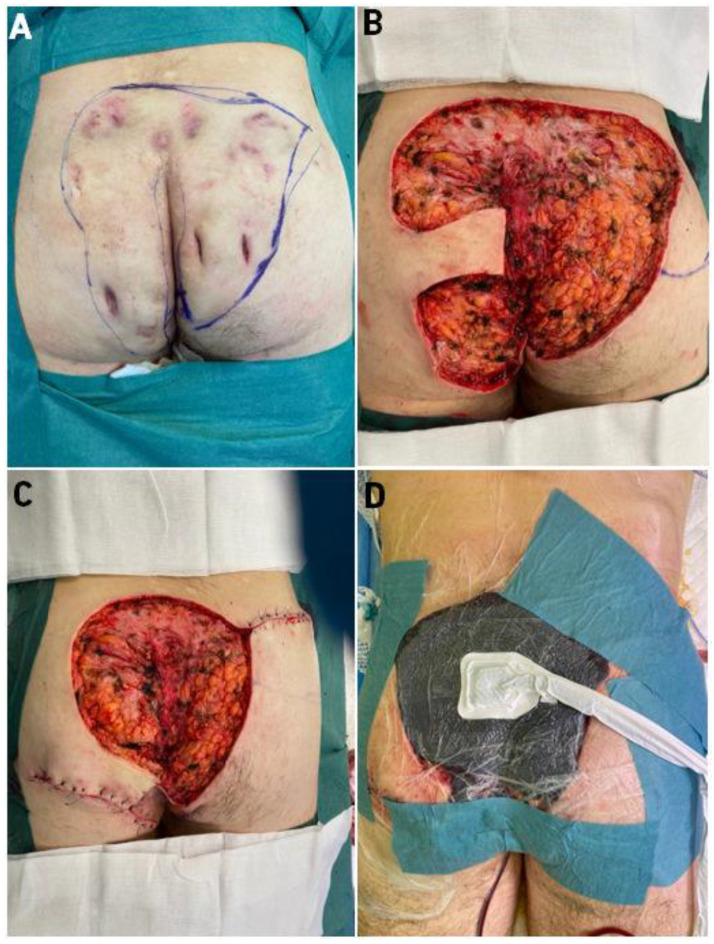
Presentation of Case 1 patient—(**A**)—HS of buttocks before the operation, excision was marked, (**B**)—immediately after excision, (**C**)—local flaps moved to partially close the wound, (**D**)—VAC therapy applied. Legend: HS—Hidradenitis Suppurativa, VAC—Vacuum-Assisted Closure.

**Figure 2 bioengineering-09-00389-f002:**
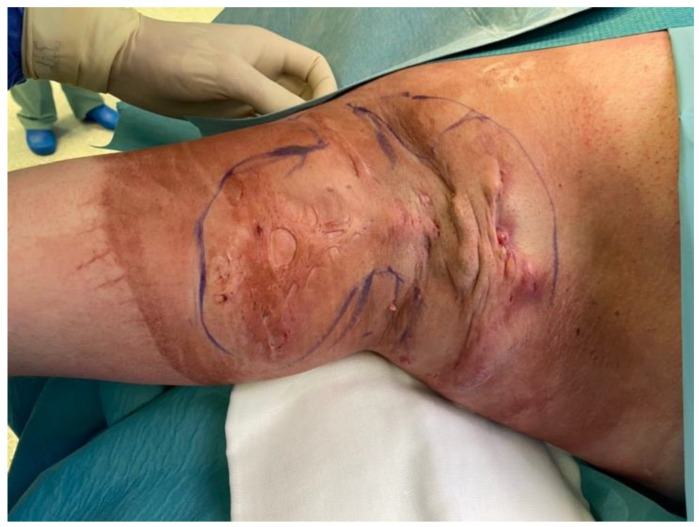
Presentation of second patient—extensive HS of right armpit, before operation. Legend: HS—Hidradenitis Suppurativa.

**Figure 3 bioengineering-09-00389-f003:**
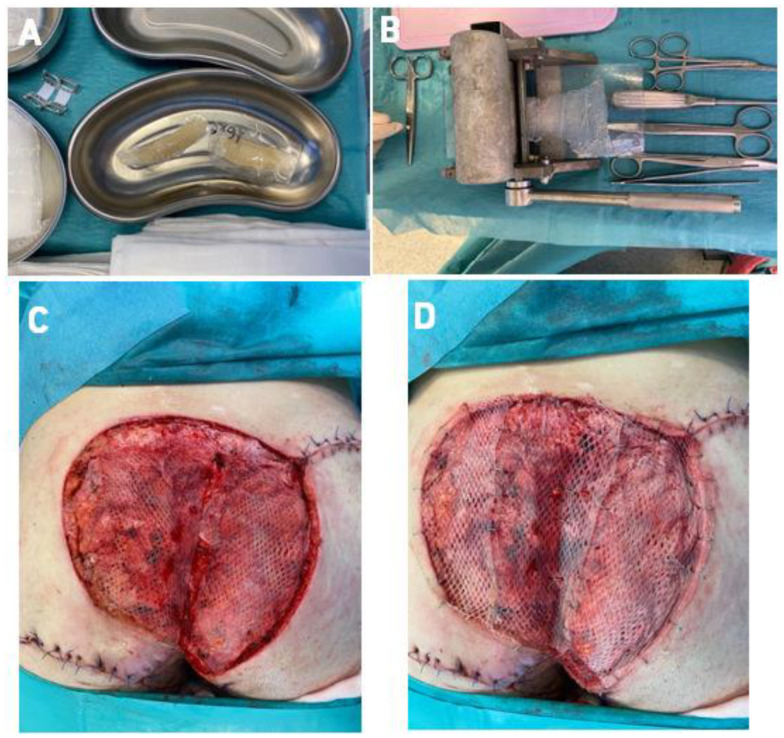
Stages of surgical procedure (case 1); (**A**)—Frozen ADM (defrosting procedure) before the operation, (**B**)—ADM meshing (ratio 1:1.5), (**C**)—application of ADM, (**D**)—STSG grafted onto ADM. Legend: ADM—Acellular Dermal Matrix, STSG—Split thickness skin graft.

**Figure 4 bioengineering-09-00389-f004:**
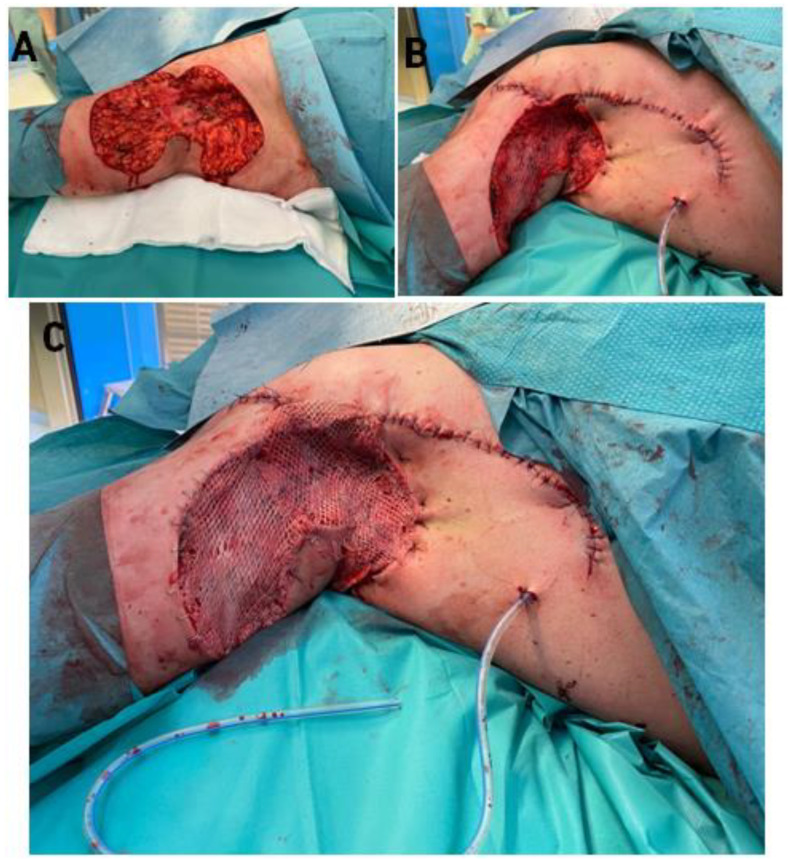
Stages of surgical procedure (case 2); (**A**)—wound after excision, (**B**)—Application of ADM directly to the wound (meshing in ration 1:2.0), (**C**)—total wound closure (co-graft ADM + STSG). Legend: ADM—Acellular Dermal Matrix, STSG—Split thickness skin graft.

**Figure 5 bioengineering-09-00389-f005:**
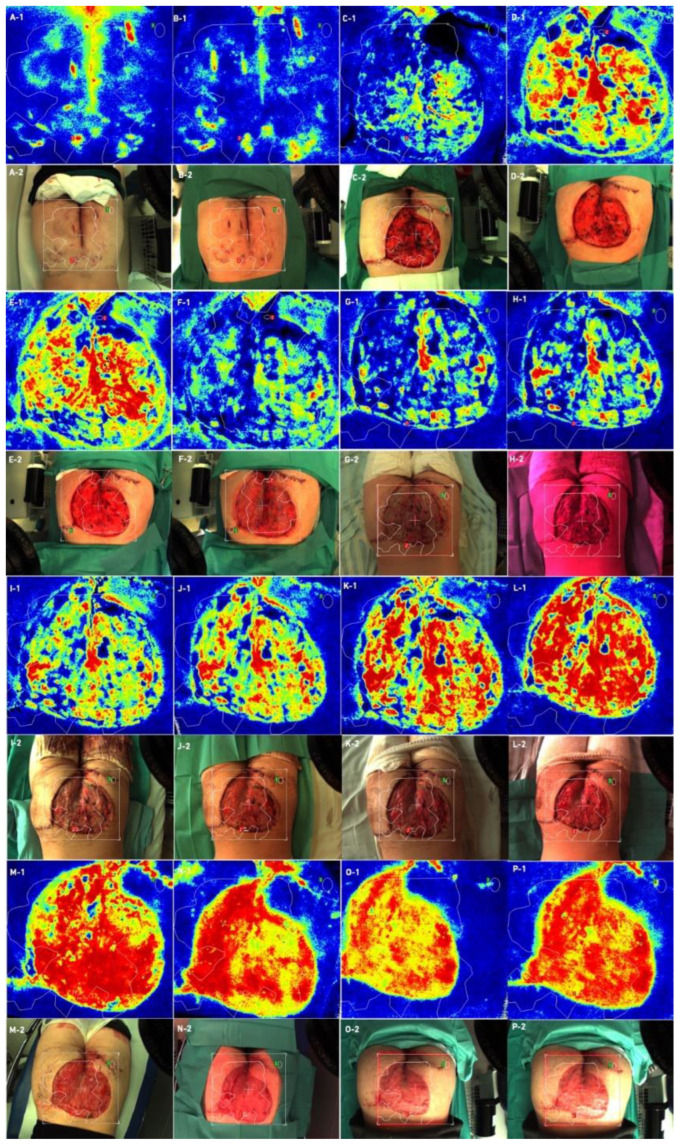
Case 1—Laser speckle analysis and wound image, (**A1**,**A2**)—Patient before admission, (**B1**,**B2**)—Patient before surgical intervention, (**C1**,**C2**)—Patient just after first surgical intervention, (**D1**,**D2**)—Patient after 5 days of VAC-therapy, (**E1**,**E2**)—Patient just before ADM and STSG application, (**F1**,**F2**)—Patient after application of ADM and STSG, (**G1**,**G2**)—Patient on third day after surgery, (**H1**,**H2**)—Patient on 5th day after surgery, (**I1**,**I2**)—Patient on 7th day after surgery, (**J1**,**J2**)—Patient on 12th day after surgery, (**K1**,**K2**)—Patient on 14th day after surgery, L(**1**,**L2**)—Patient on 16th day after surgery patient, (**M1**,**M2**)—Patient on 20th day after surgery, (**N1**,**N2**)—patient 1 month after surgery, (**O1**,**O2**)—patient 2 months after surgery, (**P1**,**P2**)—patient 2.5 months after surgery. Legend: ADM—Acellular Dermal Matrix, STSG—Split Thickness Skin Graft, VAC—Vacuum-Assisted Closure.

**Figure 6 bioengineering-09-00389-f006:**
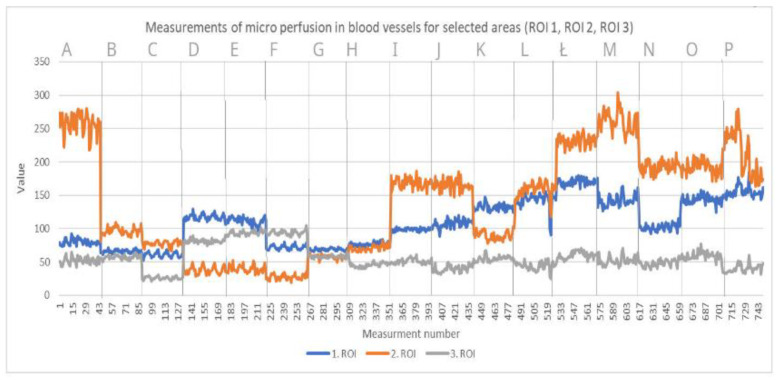
Measurements of microperfusion in blood vessels in selected areas—Case 1. Legend: ROI—Region of interest.

**Figure 7 bioengineering-09-00389-f007:**
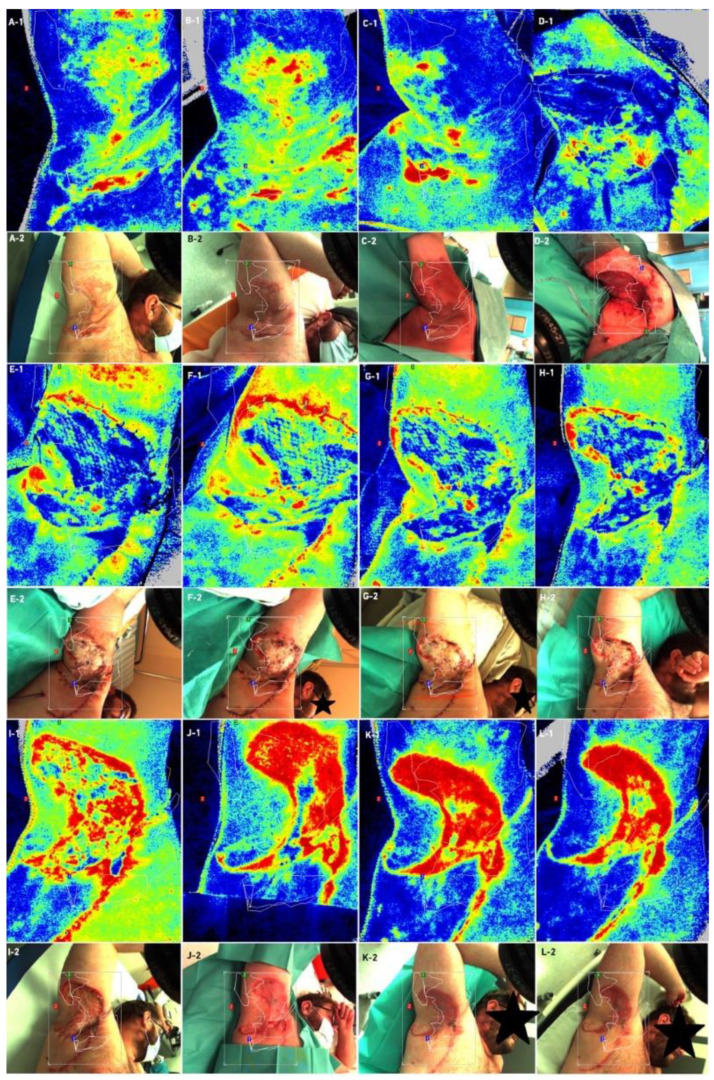
Case 2—Laser speckle analysis and wound image, (**A1**–**B2**)—Patient before admission, (**C1**,**C2**)—Patient before surgical excision, (**D1**,**D2**)—Patient after ADM and STSG grafting (**E1**,**E2**)—Patient on third day after surgery, (**F1**,**F2**)—Patient on 5th day after surgery, (**G1**,**G2**)—Patient on 7th day after surgery, (**H1**,**H2**)—Patient on 12th day after surgery, (**I1**,**I2**)—Patient on 14th day after surgery, (**J1**,**J2**)—patient 1 month after surgery, (**K1**,**K2**)—patient after 2 months after surgery, (**L1**,**L2**)—patient 2.5 months after surgery.

**Figure 8 bioengineering-09-00389-f008:**
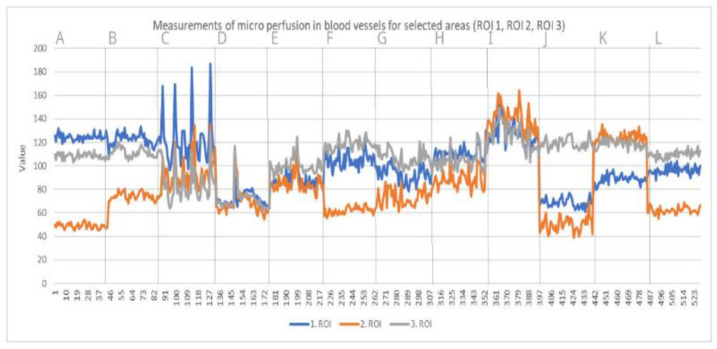
Measurements of microperfusion in blood vessels in selected areas—Case 2. Legend: ROI—Region of Interest.

**Figure 9 bioengineering-09-00389-f009:**
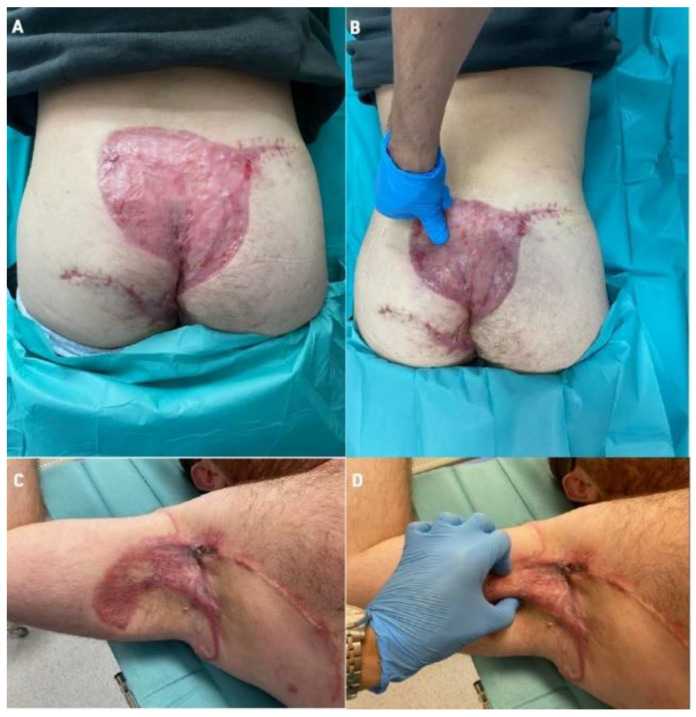
Final results 2.5 month after procedure co-graft of ADM and STSG. (**A**)—Final result of reconstructed bottocks, (**B**)—Elastic scar in the “pinch” test, (**C**)—Final result of reconstructed right axilla, (**D**)—“Pinch” test in right axilla (elastic scar). Legend: ADM—Acellular Dermal Matrix, STSG—Split thickness skin graft.

## Data Availability

Not applicable.

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
