# Peer review of "Co-Graft of Acellular Dermal Matrix and Split Thickness Skin Graft—A New Reconstructive Surgical Method in the Treatment of Hidradenitis Suppurativa"

_bioengineering, 2022, doi:10.3390/bioengineering9080389_

Round 1
Reviewer 1 Report
The manuscript describes two case of Hidradenitis (HS) treated with a combined graft in a single step surgical procedure. The patients were grafted with acellular cadaver skin prepared in house and autologous skin split skin graft. Data on blood perfusion, temperature, moisture, water loss, melanine levels, and graft thickness were collected over a period of time. Although the grafts were analysed extensively, the manuscript is low on novelty. There are a number of reports in the literature describing the allograft and split skin grafting, particularly in burns.
Specifically, I would recommend following changes:
Method- Authors need to describe the allograft processing/ preparation in method section
Results- Edit 2,5-month to 2.5-month
Supplementary data- The time points need to be reported for data collection rather than exact date.
Author Response
We would like to thank the reviewers for their insightful and valuable comments. We tried to take into account all of them and to correct the manuscript according to the recommendations. We hope that the corrections will improve the quality of the manuscript. We incorporated all the changes and marked them in the manuscript. Reviewers opinions will significantly improve this work, we have expanded the literature of the manuscript with 4 additional publications. The changed fragments in the text are underlined in yellow.
Dear Reviewer 1,
We are very grateful for all valuable suggestions, comments and remarks of the Reviewer 1. Any changes to the manuscript are highlighted in yellow in resubmitted form of this manuscript. As suggested we described the Acellular Dermal Matrix preparation in Method section. We changed 2,5 to 2.5 month. Supplementary data is corrected, we changed the figures and tables and in this form in our opinion is more readable for readers. We also changed dates to time points as Reviewer I suggested. Thank you very much for you valuable comments, it will definitely improve this manuscript.
Sincerely yours, Marcin Gierek
Reviewer 2 Report
- The abstract is poorly written and too short.
- Abbreviations should be avoided in the keywords.
- The same about the introduction
- The authors are suggested to clarify the reason for such treatment? Is there a synergistic therapeutic effect? Mention them!
- Details must be provided for instruments and materials used in the study; for example, the supplier of ADM?
- Scoring the healing process is necessary to claim the effectiveness of the applied treatment
- Mentioning the ethical approval code is mandatory
- Conclusions should be improved
Author Response
We would like to thank the reviewers for their insightful and valuable comments. We tried to take into account all of them and to correct the manuscript according to the recommendations. We hope that the corrections will improve the quality of the manuscript. We incorporated all the changes and marked them in the manuscript.Reviewers opinions will significantly improve this work, we have expanded the literature of the manuscript with 4 additional publications. The changed fragments in the text are underlined in yellow.
Dear Reviewer 2,
We are very grateful for all valuable suggestions, comments and remarks of the Reviewer 2.
Any changes to the manuscript are highlighted in yellow in resubmitted form of this manuscript.
As Reviewer 2 suggested we changed the abstract, we extend abstract in imporortant information.
Thank you for mentioning about the abbreviations in keywords – it is already corrected to full sentences. As Reviewer 2 suggested we clarify the reason of choosing this treatment in Discussion Section. We mention the synergistic therapeutic effect , Lee et al. manuscript encourage us to use this method. There is still lack of clinical trials on Co-graft. Additionally, we changed the title to “Co-graft of ADM and STSG” which in our perspectives it’s one surgery treatment (one operation). Thank you for mentioning that scoring healing process is very important. Yes, Reviewer 2 is absolutely right – we indirectly scored healing process with LASCA device (laser speckle contrast analysis) – this device in our perspectives is very accurate. Furthermore, microcirculatory is visible in LASCA device. We would like to ensure you, that in further research we will extend this valuable comment.
Institutional Ethics Committee of Silesian Medical Chamber in Katowice (protocol code 30/2021 and 22.09.2021 date of approval). Our protocol of clinical trial is also approved by Institutional Review Board in Centre Burns Treatment in Siemianowice Slaskie (No. 2/2022/CLO).
Reviewer 2 is right – we corrected conclussions, we are aware that this is pre-eliminary study based on only 2 cases, so we changed the conclussions section.
Thank you very much for you valuable comments. We are no doubt that your remarks will improve the quality of this manuscript.
Sincerely yours,
Marcin Gierek
Reviewer 3 Report
Thank you for this very interesting paper on a novel method for closure of extensive surgical wounds in patient with HS. The results are very impressive.
1. HS patients are seen by GPs, dermatologists, plastic surgeons, gastrosurgeons and in your case at a burn centre. Could you consider to shortly explain the ADM method in the introduction/methods rather than in the discussion? When discussing a method it should previously have been described. New information on methods, background or results does not belong in the discussion. Likewise other parts of the discussion belong in the introduction.
In the discussion I suggest you discuss the relevance of your results, what can be improved or how this method can help patients in the future. Also discuss longer follow up, and what may be reasons for not being able to use the method. For instance many patients with HS smoke and are obese and surgical procedures may be contraindicated because of potential complications. Strengths of the new described method, also limitations should be discussed. Generalizable for all? A real discussion is actually lacking.
You give detailed information in the supplementary material, but not on ADM. Clinicians/dermatologists who do not do extensive plastic surgery might not be familiar enough with the procedure. The paper will have a wider readershoip.It is too late to describe it in the discussion.
2. Is it common that HS patients are seen by burn centers? Would you consider to give some background for chosing to use this method in HS? Is the method used at your centre for treatment of burns or is this a completely novel approach? See also my previous remark on different specialists with different back-ground in surgical treatments.
3. You do not mention why you change the ratio for patient 2. What are the reasons, or on what grounds was ratio decided?
4. Presenting data for EQ-5D-5L: This instrument has 2 parts. The one that is scored 0-100 is the EQ-VAS. It assesses self prceived health state, but not directly quality of life. “your own health state today” on a 0 (worst imaginable) to 100 (best imaginable) scale. EQ-5D results are presentet as an index, and results are refered to as Health Related Quality of Life. The instrument does not evaluate impaired QoL due to other factors, not related to health). The results are therefore not readily understood. Please consider correcting this.
Minor comments:
Figure S14, all images: 'Gęstość skóry właściwej' Do you mean dermal density as in the above? Would you correct for the readers.
Also note that all tables and figures should be understood also when read separately. Abbreviations should be given for each table/figure in the paper and in the supplementary material.
Typos: Figure 5: You write 'M1-H2'. Do you mean M1-M2. Also description for L1-L2 and N1-N2 is missing.
Author Response
We would like to thank the reviewers for their insightful and valuable comments. We tried to take into account all of them and to correct the manuscript according to the recommendations. We hope that the corrections will improve the quality of the manuscript. We incorporated all the changes and marked them in the manuscript.Reviewers opinions will significantly improve this work, we have expanded the literature of the manuscript with 4 additional publications. The changed fragments in the text are underlined in yellow.
Dear Reviewer 3,
Thank you very much for the good evaluation of our manuscript. We are pleased that Reviewer 3 found the manuscript interesting.
Ad 1)
Thank you very much for this valuable comments and remarks. As Reviewer 3 suggested – we rebuild the manuscript. We extend the introduction section, material and methods. Reviewer 3 is right – ADM should be described well for readers in non surgical professions. We described ADM, also in material and methods we described the preparation process of ADM in our Tissue Bank Laboratory in Centre For Burns Treatment in Siemianowice Slaskie, Poland. We changed the discussion section as Reviewer 3 suggested. And Reviewer 3 is right the limitations of this method should be discussed – we change the manuscript due to the Reviewers 3 comments. We changed additionaly the title to “Co-graft of ADM and STSG”, because the former title might be confused especially for non-surgical readers. Thank you very much for your comments.
Ad 2)
Thank you very much for this comment – we appreciate this question. Maybe it’s not common in other countries, but in Poland there are 3 surgical departments in whole 38 million country. One of them performed surgeries only under local anesthesia, the second is only commercial – so not every patient can afford for this treatment. And our Department (General Surgery Department, Burn Unit in Center for Burns Treatment in Siemianowice Śląskie, Poland – where the main author is a head of the general surgery department) is the only surgical department in our district with potential HS surgical treatment.We have patients from all around the country – even 1000 km far away from our hospital. There is still lack of knowledge in the field of hidradenitis suppurativa. The median time from first symptoms to accurate diagnosis is more than 10 years. I am happy that Department I’ve been running to can help this specific patients. There is still a lot of work to do with this disease. We are very encourage of first observations with co-graft of ADM and STSG, I hope that when we will extend the research for more than 30-50 patients it will clarify that this technique will have a place in reconstructive surgery in HS. This is a novel method for us – we have Tissue Bank in our Centre so we can afford for such a treatment like ADM application. But in my opinion this is a real science in the field of reconstructive surgery – synergism of tissue engineering and surgical techniques.
Ad 3)
Thank you for this comment. You are right, we did not mention clearly why we changed the ratio. The reason of that was , first surgery was 2-3 days before the second operation. And in first 24-48 hours we saw pale STSG on ADM layer in first case. We of course done the LASCA examination – which ensure us that there is everything ok, with microperfussion/microcirculatory perfusion. And in second operation I changed the ratio to avoid the pale STSG in 24-48 hours after operation. Basically, that was the reason. Now I am after 5 surgical operations with ADM and STSG and I’m still changing the ratio of ADM and STSG. Right now I am using ratio 1:2.0 for both of them ADM and STSG – but time will tell what is the best solution for the patient. It will definitely need more further research.
Ad 4) Reviewer 3 is absolutely right. We changed in manuscript Quality of Life to health-related life quality. Of course, there are specific scales as DLQI for HS QoL assessment, but we will definitely used this scales in further research. Thank you very much for this valuable comment.
Ad minor comments :
Thank you very much for mentioning this. You are right, we changed the manuscript due to your suggestions.
We completely changed the supplementary data – in attached file there is new supplementary material. We changed supplementary data according to Reviewer 3 suggestions. We are sure that it will improve quality of this manuscript. We changed the abbreviations – we spell it out below all of the figures and tables (in manuscript and in supplementary data).
Thank you very much for this valuable comments, remarks. Thank you for your questions, I am very pleased that our manuscript opens a discussion between us. Thank you so much.
Sincerely yours,
Marcin Gierek
Reviewer 4 Report
The authors present 2 cases of treatment with Acellular Dermal Matrix and Split Thickness Skin Graft for severe hidradenitis suppurativa. Hidradenitis suppurativa sometimes occupies large areas of the body and can be a surgical challenge.
The article is well written, and the photos and graphics are good. However, from my point of view there are several weaknesses in this manuscript:
- Although the authors say that it is the first article in which this treatment is described, they have not referenced this publication: Chaffin AE, Buckley MC. Extracellular matrix graft for the surgical management of Hurley stage III hidradenitis suppurativa: a pilot case series. J Wound Care. 2020 Nov 2;29(11):624-630. doi: 10.12968/jowc.2020.29.11.624.
- It is true that the cases have been well studied, as can be seen in the supplementary material, however only two cases are too few to draw conclusions.
- The dermal matrix used is not commercial and has been produced by the authors from cadaver allograft. This can be a problem for the reproducibility of the results.
- The legends of the figures do not include the type of graph used or the units of the measurements, and if they use any statistical measure to compare the measurements (means or medians)
They should talk about the limitations, and not just implicitly as in the conclusions. The ethical aspects are not correctly presented either, because although reference is made to the Ethics committee, as it is an extracellular matrix of its own production, the design is that of a clinical trial and should be included in the methodology
Author Response
We would like to thank the reviewers for their insightful and valuable comments. We tried to take into account all of them and to correct the manuscript according to the recommendations. We hope that the corrections will improve the quality of the manuscript. We incorporated all the changes and marked them in the manuscript.Reviewers opinions will significantly improve this work, we have expanded the literature of the manuscript with 4 additional publications. The changed fragments in the text are underlined in yellow.
Dear Reviewer 4,
We are very grateful for all valuable suggestions, comments and remarks of the Reviewer 4. Any changes to the manuscript are highlighted in yellow in resubmitted form of this manuscript.
Reviewer 4 is right – we did not reference this publication. Thank you very much for mentioning this. As Reviewer 4 is suggesting we extend the references section with this Chaffin et al manuscript. However, Chaffin et al. used ADM as a tissue coverage after wide excisions – in 2 cases, and in 4 cases it was additionally added to the flap reconstructrions. In one case they grafted STSG after 22 days post ADM application. Thank you very much for mentioning this publication – it improved our discussion section. We changed title to “Co-graft of ADM and STSG” to clarify more our method, which basically is grafting ADM and STSG during one surgical treatment. So yes, Reviwer 4 is right – Chaffin et al. used first ADM as a tissue coverage after excisions, but we still hope that our co-graft technique is first paper in HS surgery. Thank you very much for this valuable comment and for finding this publication (Chaffin et al.). There is no doubt, that your remarks improves the quality of this manuscript.
Reviewer 4 is right – that ADM used in this manuscript is our in-house technique. We have Tissue Bank in our Center for Burns Treatment in Siemianowice Slaskie, Poland. The entire process of preparation of ADM is in resubmitted version of manuscript. Each ADM undergo very specific, accurate procceses with full standardization. Reviewer 4 is right it is too early to draw conclussions – we changed conclussions sections as Reviewer 4 suggested. Our further research definitely will ask the question about reproducibility of the results of ADM preparation process of our in-house Tissue Bank. Thank you very much for this valuable comment, we will concider your suggestions in further research. Your remarks will definitely improve this manuscript.
Reviewer 4 is right statistical analysis must be done. This is pre-eliminary study based on 2 cases – so it’s to early to use statistical tests. We will use statistical analysis when we reach more than 30 patients in that method. Thank you so much for mentioning this.
Reviewer 4 is right – we should talk about the limitations. We changed accordingly our discussion/conclusion section and we attached the limitations of our method. Thank you very much for this comment it will improve this manuscript.
Reviewer 4 is right – we have Ethical Committee Agreement and additionaly we asked about the permission of our Institutional Review Board in Centre For Burns Treatment. This board approved our study as a clinical trial. Code numbers are attached in resubmitted manuscript.
Thank you very much for your important comments, suggestions. There is no doubt that your valuable comments improved this manuscript, thank you so much for your remarks.
Sincerely yours,
Marcin Gierek
Round 2
Reviewer 2 Report
The authors have well addressed the proposed comments.
Reviewer 4 Report
The authors have notably improved the manuscript following suggestions from reviewers.